# Mechanical Strength of Triply Periodic Minimal Surface Lattices Subjected to Three-Point Bending

**DOI:** 10.3390/polym14142885

**Published:** 2022-07-16

**Authors:** Zo-Han Lin, Jyun-Hong Pan, Hung-Yuan Li

**Affiliations:** Department of Mold and Die Engineering, National Kaohsiung University of Science and Technology, No. 415 Jiangong Rd., Kaohsiung 807618, Taiwan; 1104406101@nkust.edu.tw (Z.-H.L.); f109147131@nkust.edu.tw (J.-H.P.)

**Keywords:** failure mode, finite element analysis, relative density, sandwich panel structure, three-point bending test, triply periodic minimal surface

## Abstract

Sandwich panel structures (SPSs) with lattice cores can considerably lower material consumption while simultaneously maintaining adequate mechanical properties. Compared with extruded lattice types, triply periodic minimal surface (TPMS) lattices have light weight but better controllable mechanical properties. In this study, the different types of TPMS lattices inside an SPS were analysed comprehensively. Each SPS comprised two face sheets and a core filled with 20×5×1 TPMS lattices. The types of TPMS lattices considered included the Schwarz primitive (SP), Scherk’s surface type 2 (S2), Schoen I-graph-wrapped package (I-WP), and Schoen face-centred cubic rhombic dodecahedron (F-RD). The finite element method was applied to determine the mechanical performance of different TPMS lattices at different relative densities inside the SPS under a three-point bending test, and the results were compared with the values calculated from analytical equations. The results showed a difference of less than 21% between the analytical and numerical results for the deformation. SP had the smallest deformation among the TPMS lattices, and F-RD can withstand the highest allowable load. Different failure modes were proposed to predict potential failure mechanisms. The results indicated that the mechanical performances of the TPMS lattices were mainly influenced by the lattice geometry and relative density.

## 1. Introduction

Triply periodic minimal surfaces (TPMSs) were originally discovered and described by Schwarz in 1865 [1]. However, fabricating these surfaces is still a challenge due to conventional technical limitations. The development of additive manufacturing (AM), particularly 3D printing, has made the fabrication of TPMSs possible. To increase the strength of AM products while reducing material consumption, the sandwich panel structure (SPS) was developed, which is a lattice-based cellular pattern enclosed by two face sheets. SPSs are widely applied in engineering and industry for aerospace, thermal systems, packaging and medical implants [2,3,4]. A wide range of lattice patterns can be used, such as honeycombs, tri-/quadri-grids, and TPMSs.

SPSs are commonly observed in nature, such as in human skulls, butterflies, and dragonflies [5]. SPSs provide an outstanding stiffness/strength-to-weight ratio and excellent energy absorption because of their very high moments of inertia [6]. When external load is applied on the face sheets, the force is dispersed to the lattice by decomposition. The force undertaken by the lattice causes the lattice structure to deform, and strain energy is absorbed in the lattice. SPSs also provide an efficient framework for resisting buckling and bending loads. Mahmoud and Elbestawi [7] found that SPSs can be used to develop artificial orthopaedic implants with a similar porosity to that of human bones. The porous structure allows bone to easily grow inside and strengthens the bond between the implant and host bone. Gibson and Ashby [3] showed that SPSs can be observed not only in biological structures and plants but also in many weight-saving products.

An SPS contains a lattice core. Several struct-based cellular geometries are possible, such as the cubic primitive, body-centred cubic, face-centred cubic, and octet truss [8,9,10]. Another existing family of geometries that offer similar advantages are TPMSs. Three major factors influence the mechanical properties of a structure: the shape, size, and relative density [11,12,13]. Cao et al. [14] conducted an experiment on P-lattice structures, which is a nature-inspired TPMS commonly seen in butterfly wings. Their results indicated that the P-lattice structure is an optimal version of the original Schwarz primitive (SP) and has exceptional mechanical properties compared with regular lattice patterns such as triangles, hexagons, and diamonds. Abueidda et al. [15] compared four TPMSs and showed that different shapes can affect the mechanical properties under compressive loading. Among the types of TPMSs considered, the Neovius surface type was found to have the highest compressive strength and absorption capability. Many studies [16,17,18] have examined the mechanical properties of a single TPMS. For a given relative density, the Schoen face-centred cubic rhombic dodecahedron (F-RD) appears to have the highest elasticity, followed by SP, Schwarz diamond, Schoen I-graph-wrapped package (I-WP), and gyroid types.

The most significant structural characteristic of a lattice pattern is the relative density ($\rho_{c}^{*}/\rho_{s}$) [19], where $\rho_{c}^{*}$ and $\rho_{s}$ are the densities of the lattice and solid materials, respectively. According to Chen et al. [20], increasing the relative density of a hybrid cellular lattice can be correlated with tremendous increases in the Young’s modulus, bulk modulus, shear modulus and total stiffness. Kladovasilakis et al. [21] tested three TPMSs at different relative densities under compressive loading, and their results revealed that SP has a higher energy absorption rate than the gyroid and Schwarz diamond for a wide range of relative density. However, the energy absorption of SP at the relative density of 10% is lower than that at 20% and 30% because of buckling on the face sheets. In addition, even though the gyroid absorbed less energy in the elastic region than SP, it still provided extraordinary energy absorption at all relative densities because of its high yield strength. Although the Schwarz diamond had the lowest energy absorption, it had the highest compressive strength. Maskery et al. [22] and Lee et al. [23] indicated that the relative density of SP can be altered by adjusting the thickness of the shell structure. Both studies concluded that the shear modulus, shear yield strength, bulk modulus, and hydrostatic yield strength scale linearly with the relative density of the SPS. The mechanical properties can be further improved by modifying the plastic compositions and processes [24,25].

Simsek et al. [26] numerically analysed different mesh types for TPMSs to determine the suitable modelling mesh. Results revealed that the shell model is appropriate for meshes with thin and uniform wall thickness. Madenci and Guven [27] and İrsel [27,28] investigated the difference between theoretical and numerical results with different meshes and showed that the difference between the theoretical and numerical analysed displacements was ~1.32% with a shell model under the same loading conditions. Alternatively, the difference was much greater at ~11.84% with a solid model. The results illustrated that shell elements are more suitable than solid elements for analysing thin to moderately thick wall thickness structures, which are characteristic of TPMSs.

TPMSs under compressive loading and sandwich beams with a 3D truss cell/foam core under three-point bending have been studied extensively [29,30,31,32]. In this study, the objective was to compare the total deformation and stress distribution of SPSs containing TPMS lattice cores with different relative densities. Both numerical and theoretical analyses were performed for the comparison. The results were used to determine the maximum allowable load before yielding and the failure mode of the SPSs.

## 2. Theoretical Background

### 2.1. Cellular Structures and Modelling Procedure of Different TPMSs

A minimal surface is defined as the surface of the minimal area between any given boundaries. When a minimal surface repeats itself by rotating, mirroring or replicating in three dimensions, it is defined as a TPMS [33]. TPMSs have complex shapes; therefore, generating these surfaces in the computer-aided design software is difficult. Mathematically, TPMSs can be expressed through different approaches, such as the parametric, implicit and boundary methods [34]. In this study, four types of TPMSs were considered, as shown in Figure 1: SP, S2, I-WP and F-RD. The implicit expressions are listed in Equations (1)–(4). All expressions are based on three coordinate variables each in the range of [–π, π] [21,22,35,36].

SP:(1)$$f_{SP}=\cos\left(x\right)+\cos\left(y\right)+\cos\left(z\right)=0$$

S2:(2)$$f_{Scherk}=\sin\left(z+\frac{\pi }{2}\right)-\sinh\left(x\right)\times \sinh\left(y\right)=0$$

I-WP:(3)$$f_{IWP}=2\times \left[\cos\left(x\right)\times \cos\left(y\right)+\cos\left(z\right)\times \cos\left(x\right)+\cos\left(y\right)\times \cos\left(z\right)\right]-\left[\cos\left(2\times x\right)+\cos\left(2\times y\right)+\cos\left(2\times z\right)\right]=0$$

F-RD:(4)$$f_{FRD}=\cos\left(x\right)\times \cos\left(y\right)\times \cos\left(z\right)-0.1\times \left[\cos\left(2x\right)\times \cos\left(2y\right)\times \cos\left(2z\right)\right]+0.1\times [\cos\left(2x\right)\times \cos\left(2y\right)\times \cos\left(2z\right)+\cos\left(2z\right)\times \cos\left(2x\right)=0$$

### 2.2. Deformation of Conventional SPSs

#### 2.2.1. Total Deformation

Various studies have performed intensive experiments on sandwich beams with extruded lattices or foam inside the core under three-point bending loads [3,37,38]. The extruded lattice typically has a triangular, quadrilateral or honeycomb structure. The foam in the core can be polymer, metallic, ceramic or glass. Gibson and Ashby’s [3] solution for the total deformation of extruded lattices and foam can be extended to this study because three-dimensional TPMSs has the same periodic and symmetric features as extruded lattices.

Deformation occurs when the SPS is loaded and reaches equilibrium. The total deformation includes the bending component $\delta_{b}$ and shear component $\delta_{s}$. Figure 2 shows a schematic of an SPS under a three-point bending load.

The total deformation, *δ*, can be calculated as follows:(5)$$\delta =\delta_{b}+\delta_{s}=\frac{pl^{3}}{B_{1}{\left(EI\right)}_{eq}}+\frac{pl}{B_{2}{\left(AG\right)}_{eq}}$$

p Load along the vertical direction.

l The length of the SPS beam.

${\left(EI\right)}_{eq}$ Equivalent flexural rigidity, as shown Equation (6).

${\left(AG\right)}_{eq}$ Equivalent shear rigidity, as shown Equation (7).

$B_{1}$ and $B_{2}$ Constants of proportionality which depend on the geometry of loading, as listed in Table 1 [3].
(6)$${\left(EI\right)}_{eq}=\frac{E_{f}bt^{3}}{6}+\frac{E_{c}^{*}bc^{3}}{12}+\frac{E_{f}btd^{2}}{2}$$
(7)$${\left(\mathrm{AG}\right)}_{eq}=\frac{bd^{2}G_{c}^{*}}{c}$$
where $E_{c}^{*}$ is the Young’s modulus of the lattice core, *E_f_* is the Young’s modulus of the solid material making up the face sheets, and $G_{c}^{*}$ is the shear modulus of the lattice core. Both $E_{c}^{*}$ and $G_{c}^{*}$ are functions of the relative density $\rho_{c}^{*}/\rho_{s}$:(8)$$E_{c}^{*}=C_{1}E_{s}{\left(\frac{\rho_{c}^{*}}{\rho_{s}}\right)}^{2}$$
(9)$$G_{c}^{*}=C_{2}E_{s}{\left(\frac{\rho_{c}^{*}}{\rho_{s}}\right)}^{2}$$
where $E_{s}$ is the Young’s modulus of the lattice material, $\rho_{c}^{*}$ is the density of the lattice core, $\rho_{s}$ is the density of the solid material and $C_{1}$ and $C_{2}$ are constants of proportionality that depend on the loading geometry, as listed in Table 1 [3].

Figure 3 shows a schematic of the relationship between the bounding box and lattice volume for an SPS. The porosity is the fraction of pore space in the bounding box and is obtained from ($1-\rho_{c}^{*}/\rho_{s}$). The relative density is given by:(10)$$Relative\;density=\frac{Lattice\;cellular\;structure\;mass/Bounding\;box\;volume\;}{Density\;of\;solid\;}$$

#### 2.2.2. Weight of the SPS

The total weight of the SPS W can be written as
(11)$$W=2\rho_{f}gblt+\rho_{c}^{*}gblc$$
where g is the standard gravity of Earth.

### 2.3. Failure Modes of Sandwich Beams

Previous studies [3,13,39] have shown that four different failure modes need to be considered for SPSs: face yielding, face wrinkling, core failure and bond failure. All these failure modes must be considered because any failure mode can occur when the geometry or the loading changes.

## 3. Numerical Approach

### 3.1. Geometry Design

In this study, Grasshopper and computer codes developed in house were used to generate single cells of TPMSs. The single cells were generated in an 8 mm × 8 mm× 8 mm cubic box and patternised to generate 20×5×1 lattices to serve as the SPS cores, as shown in Figure 4. SPSs were built with two flat layers fused to the top and bottom of each core. The overall dimensions of each SPS core were 160 mm × 40 mm × 8 mm, as given in Figure 4 and Table 2. All dimensions for the exterior geometry except the span length followed the ASTM D790 standard [40].

### 3.2. Finite Element Analysis

#### 3.2.1. Model

Many studies have demonstrated that discretising the core surface as a shell rather than a solid mesh produces superior results. This is because the thickness aspect ratio is large, and the shell model can provide more accurate results [26,27,28]. To generate the shell model, the software Rhino was used to generate TPMS geometries, which were then exported to ANSYS for finite element analysis (FEA) of the structure. The thickness of each TPMS was calculated for different relative densities, as given in Table 3. Figure 5 shows the shell models of SPSs with four different TPMS lattice cores. All numerical models are 1/2 symmetry. A vertical load of 500 N was applied to the centre of the top face sheet, and total fixed conditions are assumed at both the end supports (Figure 2).

#### 3.2.2. Materials

Acrylonitrile butadiene styrene (ABS) is a common thermoplastic polymer with nearly constant elastic modulus and other mechanical properties under the operation temperature. It has loads of advantages, such as great heat resistance, high tensile strength, and low cost. Hence, ABS was selected as the material of the SPS specimens, and Table 4 presents its mechanical properties.

## 4. Results and Discussion

In this study, the mechanical performances of SPSs with four different TPMS lattice cores were evaluated and compared. In addition, the calculation results based on the equations presented in the ‘Theoretical background’ section were compared with the numerical results obtained using the FEA approach.

### 4.1. Weight Difference

To compare the mechanical performance of SPS with different TPMS lattices, the total weight of each SPS must be considered first. The weight calculated from Equation (11) and the weight acquired from the FEA models are listed in Table 5. The difference was <3% for all SPSs, which indicates that the geometry of the FEA model was sufficiently accurate and that Equation (11) could precisely calculate the weight of the SPS. Furthermore, Table 5 indicates that the increase in the relative density can increase the total weight of the SPS.

### 4.2. Deformation

The total deformations of the SPSs with different TPMS lattices cores under a three-point bending load calculated using Equation (5) were compared against the numerical predictions obtained by FEA. Table 6 lists the calculated parameters, maximum total deformation and the numerical predictions. The difference between the calculated results and numerical predictions was 1%–21% at all relative densities, which indicates good agreement. For the SP lattice core SPS, the calculated values were 1.6%–17.5% higher than the numerical prediction. F-RD lattice core SPS has the largest difference of approximately 20.7% at 10% relative density. It is because the F-RD lattice is the most complex structure compared to other types of TPMS in this study. The meshing is highly complex on the connecting mesh boundary, and the thickness of the F-RD lattice with 10% relative density is the thinnest. This is considered to be the reason behind this highest difference. Figure 6 presents the comparison between the calculated and predicted maximum deformations of SPSs with different TPMS lattice cores. The deformation decreases with increasing relative density.

If the S2 lattice cell rotates from its original array direction (Figure 7a) along the vertical axis (Figure 7b), the numerical prediction of the maximum deformation at different relative densities will be altered. S2 with the rotated direction is designated as S2_90_ in the followings. Table 7 and Figure 8 show that the numerical results of maximum deformations of S2_90_ at relative densities of 10% and 20% were extremely large compared to the calculated results. This clearly indicates that the cell direction causes this extremely huge difference. Other TPMS lattices considered in this study (i.e., SP, I-WP and F-RD) do not demonstrate this difference because their cell rotation does not affect the array direction (Figure 4). In addition, the maximum deformation decreased as the relative density increased.

The above results show that the calculation method and parameters used for the total deformation are not only adequate for TPMS structures but also can be used to predict the structures’ mechanical performance. Moreover, the cell direction appears to play an important role in the mechanical performance of the SPS with the S2 lattice core. Generally, when the thickness of wall increases, which will increase the relative density simultaneously, the weight of the SPS will correspondingly be increased and the total deformation will decrease.

### 4.3. Comparison of Maximum Deformation of Different TPMS Lattice Cores in the SPS

Figure 9 shows the deformation distribution of SPSs with different TPMS lattice cores under three-point bending. The maximum deformation was at the mid-span and possessed symmetry along the span. Therefore, the maximum deformation at the centre of the SPS beam was considered to represent the total deformation. Figure 10 shows the numerical prediction of the maximum deformations of the different TPMS lattices under three-point bending. At the same loading force and relative density, the SP lattice resulted in the smallest deformation. At the relative density of 10%, SP had the smallest deformation, followed by S2, I-WP and F-RD; at 20% relative density, SP had the smallest deformation, followed by S2, F-RD, and I-WP; as the relative density reached 30%, SP still maintained the smallest deformation, followed by F-RD, S2 and I-WP. In general, the thicker the lattice is, the stronger the SPS structure will be. The S2 lattice possesses the maximum thickness in this study. However, the S2 lattice structure has many discontinuities on its geometry boundary, and the shape is not as normalised as the SP surface. This gives rise to that the stiffness of the SPS with SP lattices core is higher than the S2 lattice, and its maximum deformation is the lowest when compared to the other types of TPMS. Moreover, the I-WP lattice provides the lowest stiffness, even when the relative density is increased from 20% to 30%. Results also indicate that the lattice geometry is a crucial factor for the mechanical performance. Furthermore, Figure 10 also demonstrates that when the relative density increases, the maximum deformation correspondingly decreases.

Because the numerical results were based on a three-point bending load, the deformation in the loading direction was also considered. Table 8 and Figure 11 compare the *Z*-axis deformations of the SPSs with different lattice cores. F-RD resulted in the smallest *Z*-axis deformation, which is because the lattice was being twisted when subjected to bending. The linear approximations of the total deformation error estimation, R^2^, of the SPSs with different TPMS lattice cores versus the relative density are presented in Table 9 and Figure 10. R^2^ was close to 1 for all SPSs with the different lattice cores, which indicates that the linear approximated total deformation from Equations (12)–(15) can reliably predict the deformations of SPSs with different TPMS lattices.

### 4.4. Comparison of Maximum Allowable Load of Different TPMS Lattice Cores in the SPS

The SPSs were made from ABS, and its maximum yield strength is about 45 MPa. Thus, the maximum allowable load at different relative densities can be calculated by numerical analysis. Figure 12 shows the results for SPSs with different TPMS lattice cores. At relative densities of 10% and 20%, SP had the highest maximum allowable loads of 584 and 702 N, respectively. However, at a relative density of 30%, F-RD had the highest maximum allowable load of 790 N, followed by I-WP, SP and S2. The maximum allowable load increased with the relative density. These results are in good agreement with those of Kladovasilakis et al. [21]. The cell direction of the S2 lattice core was again found to have a significant influence. For the S2 lattice, the maximum allowable load consistently increased with the relative density. In contrast, the maximum allowable load for the S2_90_ lattice did not increase when the relative density was increased from 20% to 30% but instead declined. In addition, the maximum allowable load was generally lower for S2_90_ than for S2. These results demonstrate that the geometry and relative density of the lattice core are significant to the structural performance. The maximum allowable load increased with the relative density. The array direction of the lattice had a significant effect on the performance of S2, which was the only TPMS in this research with different cell direction because of its two openings in the geometry.

Table 10 lists the von Mises stresses of the four TPMSs. Figure 13 shows the stress distribution of the SP lattice core inside the SPS. The maximum stress was at the interface between the lattice and face sheets. The maximum yield strength of the SPSs was about 45 MPa. Thus, the von Mises stress exceeded the yield strength for some of the TPMSs, and plastic deformation occurred. Increasing the relative density decreased the von Mises stress and improved the stiffness of the SPS.

### 4.5. Failure Mode

An SPS beam may fail according to several failure modes. All failure modes must be considered to determine which is dominant. With regard to face yielding, plastic deformation occurred once the stress on the face sheet was greater than the yield strength of the face material. The face normal stress $\sigma_{f}$ was calculated from Equation (16) to be about 62.5 MPa. This value is greater than the yield strength of the material, which means that face yielding would occur regardless of the TPMS lattice core inside the SPS.

The shear stress has the largest effect on the core tearing during the three-point bending test [11,29,30]. The shear strength of ABS is taken as half of the yielding stress. Table 11 presents the shear stress τ at the SPS core predicted by FEA. For all TPMS lattice cores, yielding occurred at a relative density of 10%. When the relative density was increased to 20%, only the F-RD lattice core yielded. Face wrinkling was only observed for the SPS with an S2 lattice core at a relative density of 10% relative density from Table 11. The bond stress $\sigma_{b}$ was calculated from Equation (17), and the results are presented in Table 12. Bond failure was highly likely for all SPSs considered in this study, because the bond stresses exceeded the yield strength.

In conclusion, the normal failure modes of the SPSs were face yielding and bond failure because of the thinness of the face sheets. Between these two failure modes, face yielding was the most severe because the stress greatly surpassed the yield strength of the material. Other failure modes, such as face wrinkling and core failure, can be prevented by increasing the relative density. These results agree with those of other studies [26,37] showing that increasing the thickness of the face sheets or using a tougher material can decrease the face normal stress and bond stress, which can prevent both failure modes.

Face stress:(16)$$\sigma_{f}=\frac{Pl}{4btc}$$

Bond stress:(17)$$\sigma_{b}={\left(G\times \frac{E_{f}}{t}\right)}^{1/2}$$

Strain energy release rate:(18)$$G=\frac{M^{2}}{2b{\left(EI\right)}_{eq}}$$

Moment:(19)$$M=\frac{Pl}{B_{3}}$$

## 5. Conclusions

In this study, the mechanical performances of SPSs with four types of TPMS lattice cores at different relative densities were evaluated thoroughly. FEA results for the deformation under the three-point bending test showed good agreement with the calculated values, with a difference of less than 21%. The lattice geometry and relative density were the primary factors affecting the mechanical performance. For a fixed relative density, the SPS with the SP lattice core had the smallest deformation, and the F-RD lattice core had the maximum allowable load before yielding. The cell direction had a significant effect on the mechanical performance of the SPS with the S2 lattice core. The most dominant failure mode was face yielding because the stress greatly surpassed the yield strength of the material used in this study. Other failure modes such as face wrinkling and core failure can be prevented by increasing the relative density. Increasing the total amount of the lattice core or using different materials for the face sheets can improve the mechanical performance of SPSs, and the SPS can be optimised to meet the requirements for various engineering applications.

## Figures and Tables

**Figure 1 polymers-14-02885-f001:**
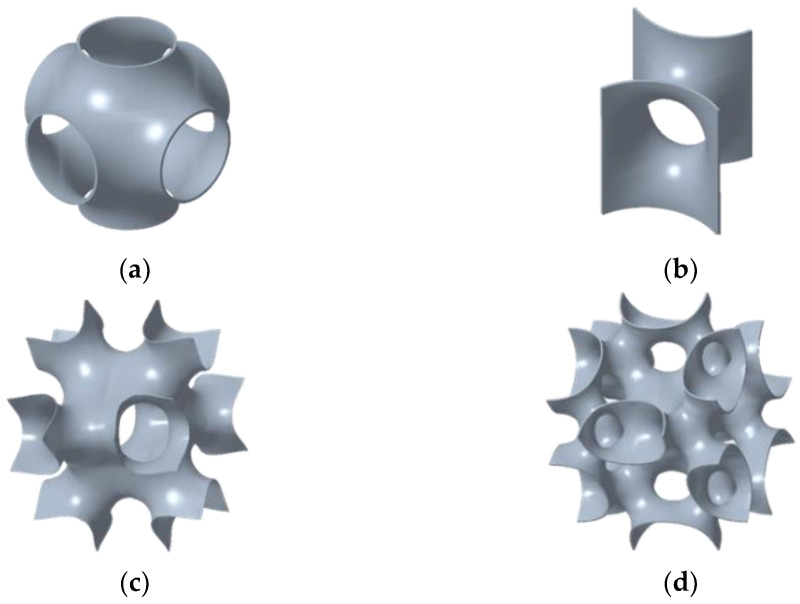
Cell geometries of different TPMSs: (**a**) SP, (**b**) S2, (**c**) I-WP and (**d**) F-RD.

**Figure 2 polymers-14-02885-f002:**
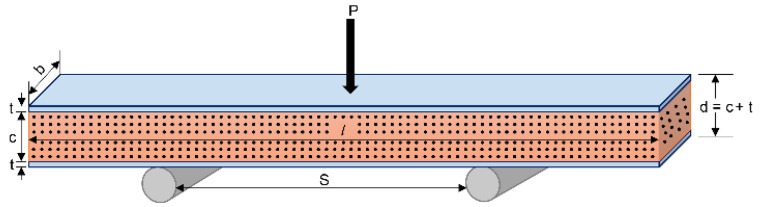
SPS under a three-point bending load.

**Figure 3 polymers-14-02885-f003:**
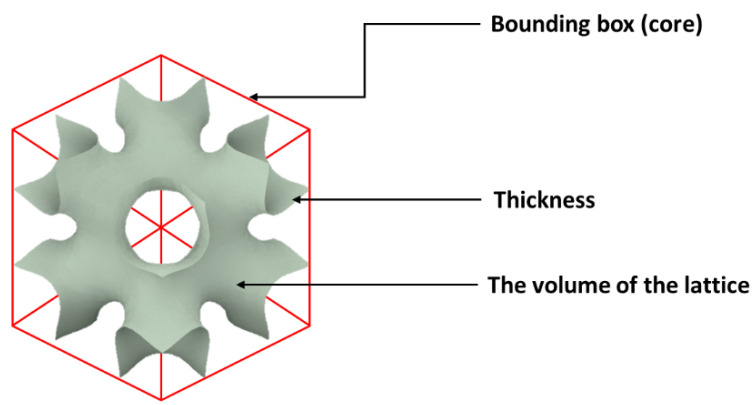
Schematic of the bounding box and lattice volume of an SPS.

**Figure 4 polymers-14-02885-f004:**
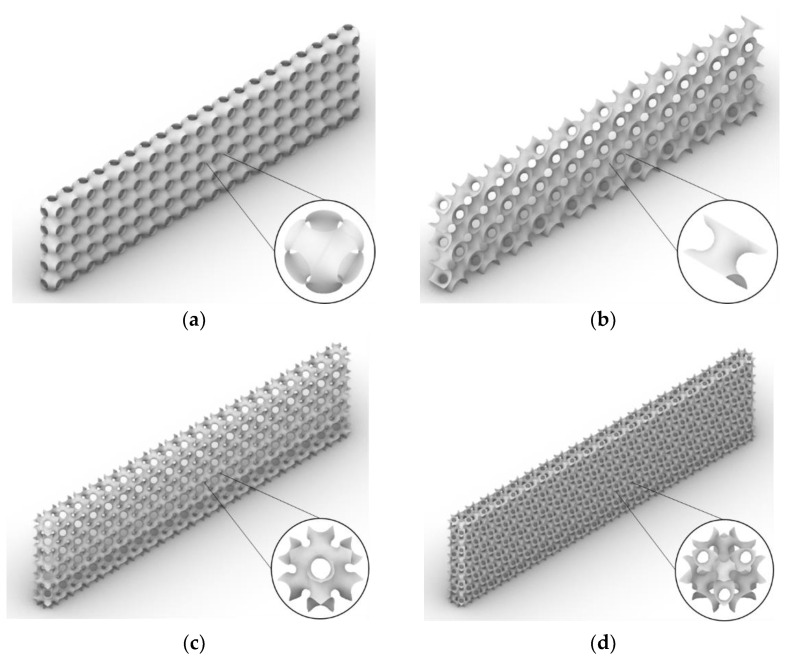
Patterns used to generate SPSs with different TPMS lattice cores: (**a**) SP, (**b**) S2, (**c**) I-WP and (**d**) F-RD.

**Figure 5 polymers-14-02885-f005:**
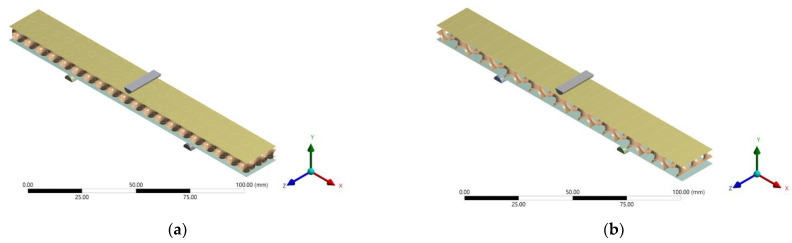
SPS models with different TPMS lattice cores: (**a**) SP, (**b**) S2, (**c**) I-WP and (**d**) F-RD.

**Figure 6 polymers-14-02885-f006:**
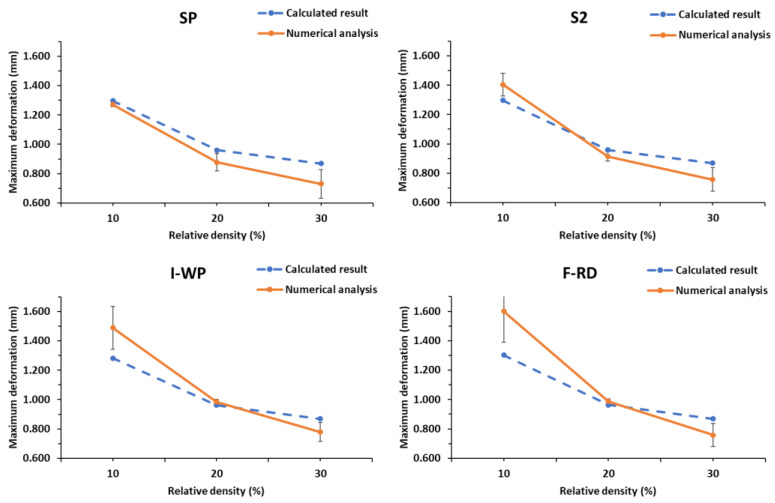
Comparison of maximum deformations between the calculation results and numerical predictions.

**Figure 7 polymers-14-02885-f007:**
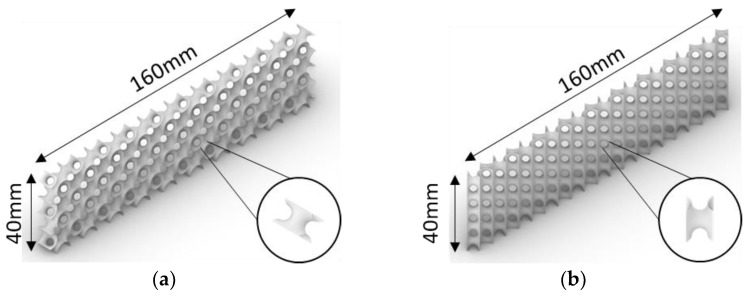
Different cell directions of the S2 lattice core: (**a**) original and (**b**) vertical direction (i.e., S2_90_).

**Figure 8 polymers-14-02885-f008:**
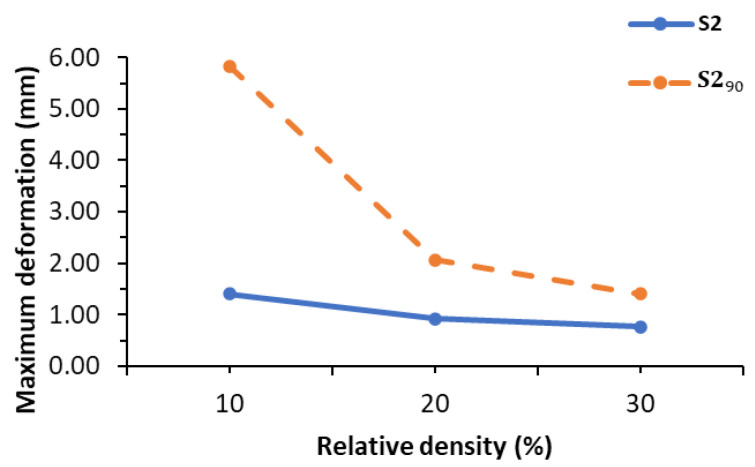
Maximum deformation results of S2 and S2_90_.

**Figure 9 polymers-14-02885-f009:**
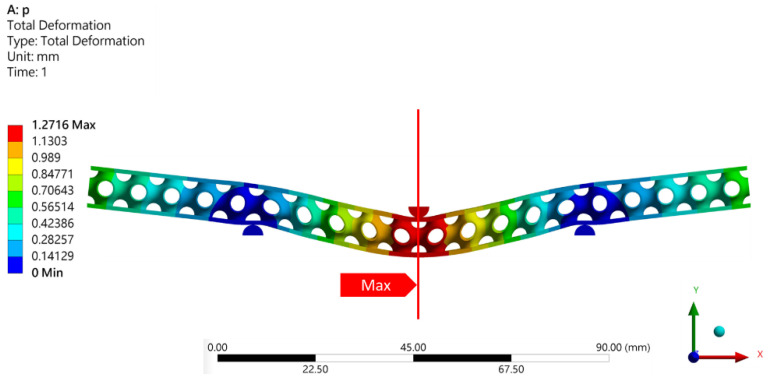
Deformation distribution of SPSs with SP lattice cores.

**Figure 10 polymers-14-02885-f010:**
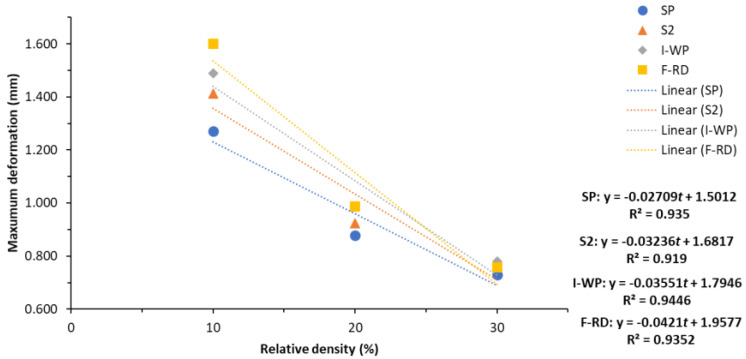
Maximum deformations of SPSs with different TPMS lattice cores.

**Figure 11 polymers-14-02885-f011:**
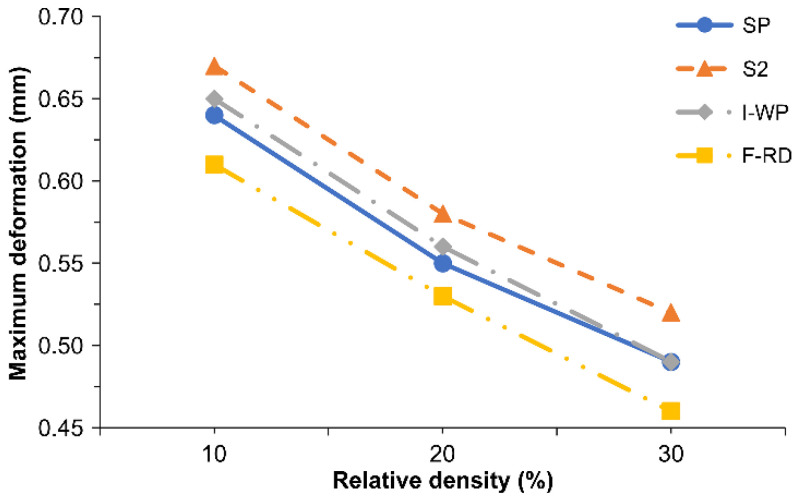
*Z*-axis deformations of SPSs with different TPMS lattice cores.

**Figure 12 polymers-14-02885-f012:**
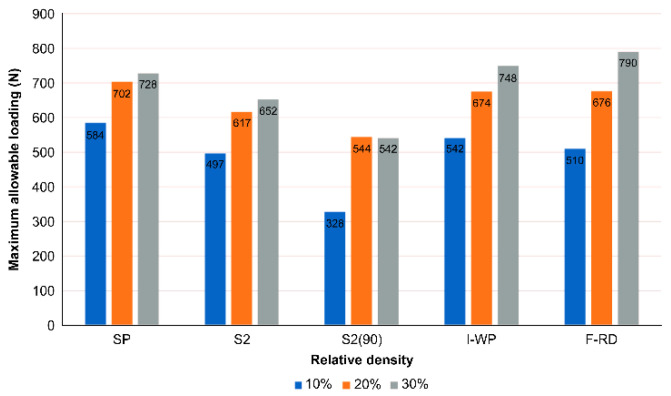
Maximum allowable loads of four TPMSs at different relative densities.

**Figure 13 polymers-14-02885-f013:**
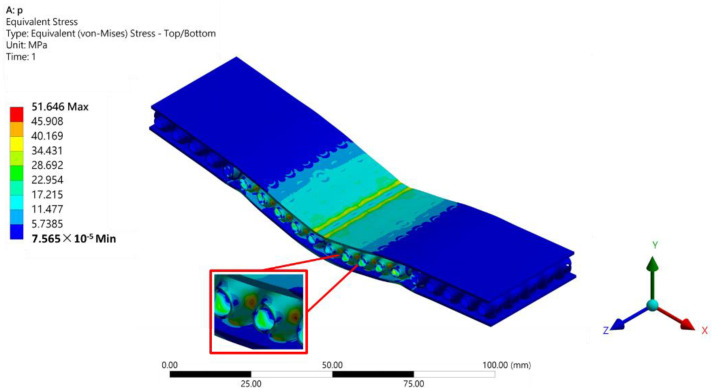
Stress distribution of the SP lattice core inside the SPS.

**Table 1 polymers-14-02885-t001:** Constants for the three-point bending testing [3].

Loading Mode	$C_{1}$	$C_{2}$	$B_{1}$	$B_{2}$
	$E_{c}^{*}=C_{1}E_{s}{\left(\frac{\rho_{c}^{*}}{\rho_{s}}\right)}^{2}$	$G_{c}^{*}=C_{2}E_{s}{\left(\frac{\rho_{c}^{*}}{\rho_{s}}\right)}^{2}$	$\delta_{b}=\frac{pl^{3}}{B_{1}{\left(EI\right)}_{eq}}$	$\delta_{s}=\frac{pl}{B_{2}{\left(AG\right)}_{eq}}$
Three-point bending, loading at the centre	1	0.4	48	8

**Table 2 polymers-14-02885-t002:** SPS parameters.

Geometric Parameter	Definition	Value	Unit
b *	Width of the sandwich beam	40	mm
c	Core thickness of the sandwich beam	8	mm
d	Distance between centroids of the upper and lower face sheets	9	mm
l	Total length of the sandwich beam	160	mm
S	Span length	80	mm
t	Thickness of a face sheet	1	mm

* Refer to Figure 2 for the definitions of the geometric parameters.

**Table 3 polymers-14-02885-t003:** Wall thicknesses of TPMS lattices at different relative densities.

Core Type	Thickness (mm)		
	10%	20%	30%
SP	0.34	0.68	1.00
S2	0.44	0.90	1.34
I-WP	0.23	0.45	0.68
F-RD	0.17	0.33	0.50

**Table 4 polymers-14-02885-t004:** Mechanical properties of ABS [41].

Property	Face Sheets and Core
$\mathrm{Density}\;\left(\mathrm{kg}/m^{3}\right)$	1040
Young’s modulus (GPa)	2.39
Poisson’s ratio	0.39
Tensile yield strength (MPa)	41.4
Tensile ultimate strength (MPa)	44.3

**Table 5 polymers-14-02885-t005:** Differences between the calculated weight and numerical prediction for each SPS.

Lattice Type	Relative Density	Calculated Weight	Weight Acquired by FEA	Difference
	(%)	(g)	(g)	(%)
SP	10	18.3	18.6	1.6
	20	23.5	23.9	1.7
	30	28.4	28.9	1.7
S2	10	18.1	18.5	2.2
	20	23.5	24.0	2.1
	30	28.6	29.2	2.1
I-WP	10	18.3	18.7	2.2
	20	23.4	23.8	1.7
	30	28.7	29.2	1.7
F-RD	10	18.3	18.7	2.2
	20	23.4	23.9	2.1
	30	28.7	29.3	2.1

**Table 6 polymers-14-02885-t006:** Differences between the calculated weight and numerical prediction for each SPS.

Lattice	Relative Density (%)	$E_{c}^{*}\;(\mathrm{MPa})$	$G_{c}^{*}\;(\mathrm{MPa})$	EI (MPa)	AG (GPa)	Calculated $\delta_{E}\;(\mathrm{mm})$	FEA $\delta_{FE,\;max}\;(\mathrm{mm})$	Difference (%)
SP	10	23.95	9.57	4.86	4.79	1.29	1.27	1.6
	20	95.79	38.32	4.98	19.15	0.96	0.88	8.7
	30	216.00	86.40	5.19	43.20	0.87	0.73	17.5
S2	10	23.21	9.28	4.86	4.79	1.29	1.41	8.9
	20	97.11	38.84	4.98	19.42	0.96	0.92	4.3
	30	215.29	86.12	5.18	43.06	0.87	0.77	12.2
I-WP	10	24.73	9.89	4.86	4.95	1.28	1.49	15.2
	20	94.67	37.87	4.98	18.93	0.96	0.99	3.1
	30	216.18	86.47	5.19	43.24	0.87	0.78	10.9
F-RD	10	24.00	9.60	4.82	4.80	1.30	1.60	20.7
	20	94.19	37.67	4.98	18.84	0.96	0.99	3.1
	30	216.23	86.49	5.19	43.25	0.87	0.75	14.8

**Table 7 polymers-14-02885-t007:** Percentage difference between S2 and S2_90_.

Relative Density (%)	S2	S2_90_	Difference between S2 and S2_90_ (%)
	$\delta_{FE,max}\;(\mathrm{mm})$	$\delta_{FE,max}\;(\mathrm{mm})$	
10	1.41	5.83	313.5
20	0.92	2.07	125.0
30	0.77	0.76	83.1

**Table 8 polymers-14-02885-t008:** Deformation along the *Z*-axis.

Lattice	Relative Density (%)		
	10	20	30
SP	0.64 mm	0.55 mm	0.49 mm
S2	0.67 mm	0.58 mm	0.52 mm
I-WP	0.65 mm	0.56 mm	0.49 mm
F-RD	0.61 mm	0.53 mm	0.46 mm

**Table 9 polymers-14-02885-t009:** Linear approximations of the deformation.

Lattice	Equation	
SP	$\delta_{SP}=-0.2709t+1.5012;\;R^{2}=0.935$	(12)
S2	$\delta_{ST}=-0.3236t+1.6817;\;R^{2}=0.919$	(13)
I-WP	$\delta_{IWP}=-0.3551t+1.7946;\;R^{2}=0.945$	(14)
F-RD	$\delta_{FRD}=-0.4214t+1.9577;\;R^{2}=0.935$	(15)

**Table 10 polymers-14-02885-t010:** von Mises stresses of the four TPMS lattices at different relative densities.

Lattice Pattern	Relative Density (%)	Von Mises Stress (MPa)
SP	10	51.65
	20	33.77
	30	30.88
S2	10	67.59
	20	42.96
	30	34.50
I-WP	10	86.46
	20	57.27
	30	45.21
F-RD	10	151.16
	20	83.96
	30	54.87

**Table 11 polymers-14-02885-t011:** Values of face normal stress under different relative densities.

Lattice Pattern	Failure Mode	Relative Density (%)		
		10	20	30
SP	$\sigma_{fw}$ (MPa)	38.48	32.05	30.88
	$\tau_{cf}$ (MPa)	27.78	12.74	8.71
S2	$\sigma_{fw}$ (MPa)	45.17	36.42	34.50
	$\tau_{cf}$ (MPa)	27.53	11.97	9.28
I-WP	$\sigma_{fw}$ (MPa)	41.51	33.35	30.07
	$\tau_{cf}$ (MPa)	31.23	17.75	13.45
F-RD	$\sigma_{fw}$ (MPa)	43.89	33.27	28.41
	$\tau_{cf}$ (MPa)	31.23	26.19	16.04

**Table 12 polymers-14-02885-t012:** Maximum bond stresses.

Lattice Pattern		Relative Density (%)		
		10	20	30
SP	$G\;(J/m^{2}$)	1029.5	1004.1	964.4
	$\sigma_{bf}$ (MPa)	49.71	49.09	48.11
S2	$G\;(J/m^{2}$)	1029.5	1003.7	964.6
	$\sigma_{bf}$ (MPa)	49.71	49.08	48.12
I-WP	$G\;(J/m^{2}$)	1029.2	1004.5	964.6
	$\sigma_{bf}$ (MPa)	49.70	49.10	48.11
F-RD	$G\;(J/m^{2}$)	1038.2	1004.7	964.3
	$\sigma_{bf}$ (MPa)	49.92	49.10	48.11

## Data Availability

The data presented in this study are available on request from the corresponding author.

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
