# Peer review of "Mechanical Strength of Triply Periodic Minimal Surface Lattices Subjected to Three-Point Bending"

_polymers, 2022, doi:10.3390/polym14142885_

Round 1

Reviewer 1 Report

1. Is the model described in this manuscript suitable for polymer materials

2. It is true that the sandwich structure can improve the impact resistance of the material, but how is the force between the sandwich interfaces dispersed in the presence of the lattice

3. Since shape and size affect the mechanical properties of materials, whether the comparison of mechanical properties in this manuscript adopts a consistent benchmark

4. Is Equation 8 correctModulus is generally linearly related to density.

5. The deformation decreases as the density increases. Why does SP have the smallest deformation behavior when the relative density is 10%

Author Response

2022/07/05

Prof. Dr. Alexander Böker

Editor-in-Chief

Polymers

Manuscript ID: polymers-1795329

Manuscript title: Mechanical strength of triply periodic minimal surface lattices subjected to three-point bending

Authors: Zo-Han Lin et al.

Dear Editor,

Thank you for your letter dated on 07/01/2022. We are pleased to know that our manuscript has been rated as potentially acceptable for publication in Polymers, subject to adequate revision and response to the comments raised by the reviewers.

Based on the instructions provided in the decision letter and comments provided by the reviewers, we have revised the manuscript by modifying the relevant sections. We have uploaded a copy of the original manuscript marked with all the changes made during the revision process. The revisions made are shown in tracks. Also appended to this letter are point-by-point responses to the comments raised by the reviewers.

We would like to take this opportunity to express our sincere thanks to the reviewers who identified areas of the manuscript that needed corrections or modification. We would like also to thank you for allowing us to resubmit a revised copy of the manuscript.

We hope that the revised manuscript is accepted for publication in Polymers.

Sincerely Yours,

Hung-Yuan Li

Department of Mold and Die Engineering,

National Kaohsiung University of Science and Technology,

No.415 Jiangong Rd., Kaohsiung 807618, Taiwan, ROC

Phone No: +886-922-852-877

Fax No: +886-7-3925469

Email Address: [email protected]

                        [email protected]

Reviewer 2 Report

The manuscript is interesting and it was correct prepared therefore I can suggest be accepted after minor revision.

The following are a few comments.

 1. Why was there as high as 21% difference between numerical and analytical models? The authors should explain this in the discussion and conclusion.

 2. Show R-square and regression equation on the correlation curves.

 3. Add error bars to all curves as well as bar charts.

 4. What is the source of data on Table 4. Was it from literature? provide reference. If it was measured, what is the standard deviation?

 5. Why was acrylonitrile butadiene styrene selected?

6. Please explain the rationale to study 3-point bending instead of 4-point bending.

7- Give details of the boundary conditions.

Author Response

(The authors gave the same response as above.)

Reviewer 3 Report

Journal of Polymers 

Technical, grammatical, and common mistakes are as follows

Comments for authors;

Ø Write keywords in alphabetical order.

Ø Make bar and border bold of all figures.

Ø %, °C, abbreviations, Figure, Table, Heading, Sub-heading, Numbers, etc., write the same format throughout the manuscript.

Ø Figures 10, and 11. Explain with more explanation.

Ø Use EndNote or Mendeley for reference the present format is not according to the prestigious journal.

§    Cite the latest some of the mentioned references.

v   https://doi.org/10.1002/app.51191

https://doi.org/10.1002/app.50515

Author Response

(The authors gave the same response as above.)
